# An Evolutionary Perspective on Hox Binding Site Preferences in Two Different Tissues

**DOI:** 10.3390/jdb9040057

**Published:** 2021-12-13

**Authors:** Laura Folkendt, Ingrid Lohmann, Katrin Domsch

**Affiliations:** 1Developmental Biology, Erlangen-Nürnberg University, 91058 Erlangen, Germany; laura.folkendt@web.de; 2Centre for Organismal Studies (COS) Heidelberg, Heidelberg University, 69120 Heidelberg, Germany

**Keywords:** Hox genes, Ultrabithorax (Ubx), binding specificity, mesoderm, evolution

## Abstract

Transcription factor (TF) networks define the precise development of multicellular organisms. While many studies focused on TFs expressed in specific cell types to elucidate their contribution to cell specification and differentiation, it is less understood how broadly expressed TFs perform their precise functions in the different cellular contexts. To uncover differences that could explain tissue-specific functions of such TFs, we analyzed here genomic chromatin interactions of the broadly expressed *Drosophila* Hox TF Ultrabithorax (Ubx) in the mesodermal and neuronal tissues using bioinformatics. Our investigations showed that Ubx preferentially interacts with multiple yet tissue-specific chromatin sites in putative regulatory regions of genes in both tissues. Importantly, we found the classical Hox/Ubx DNA binding motif to be enriched only among the neuronal Ubx chromatin interactions, whereas a novel Ubx-like motif with rather low predicted Hox affinities was identified among the regions bound by Ubx in the mesoderm. Finally, our analysis revealed that tissues-specific Ubx chromatin sites are also different with regards to the distribution of active and repressive histone marks. Based on our data, we propose that the tissue-related differences in Ubx binding behavior could be a result of the emergence of the mesoderm as a new germ layer in triploblastic animals, which might have required the Hox TFs to relax their binding specificity.

## 1. Introduction

The development of multicellular organisms is coordinated by distinct sets of transcription factors (TFs) that work in networks to initiate cell fate decision, specification, cell localization and tissue differentiation [1,2,3]. TFs functions are mostly defined by their expression patterns, which can be cell-type specific, broad, or ubiquitous, and by their mode of action as activators, repressors, or modulators. Cell-type specific TFs accomplish their function in a highly specific manner restricted to defined cell types, whereas broadly and ubiquitous TFs function in multiple and different cell types at the same time to regulate various regulatory networks [4]. Even through the function of cell-type specific TFs is under constant investigation and thus rather well established [1,5,6,7], it is less understood how broadly expressed TFs perform their diverse yet precise functions in different cell and tissue types. To dissect the specificity of a broad TF, the *Drosophila* Hox TFs provide an ideal model.

*Hox* genes are evolutionary conserved and specify the segment identity along the anterior-posterior axis of the developing embryo [8,9,10,11]. Thus, they are required in the different cell and tissue types of a defined segment. Despite their highly specific function in vivo, Hox proteins are characterized by a rather unspecific binding behavior in vitro, a phenomenon known as the Hox paradox [12,13,14]. For decades, this paradox has been intensively investigated, which resulted in deep insights from in vitro experiments enriching for DNA sequences with which Hox proteins (and their cofactors) interact [15], or from in vivo studies testing the binding abilities of different Hox proteins to selected (ectodermal/neuronal) enhancers [16,17]. All of these studies led to several working models explaining how Hox TFs active in different body parts perform specific functions in each segment. One of the most prominent models is that regional specificity in the interaction of different Hox proteins with their regulatory regions is achieved via Hox-specific low-affinity binding sites [16]. Despite these major advancements, several questions with regards to Hox specificity are still unsolved. In particular, it is completely unclear how one Hox protein can act with such high specificity in the different cell and tissue types within one segment.

To contribute to the field and tackle this question in a global fashion, we used the Hox TF Ultrabithorax (Ubx) and bioinformatic approaches to analyze the binding behavior and preferences of Ubx in two tissues, the mesoderm and the neuronal system. These investigations showed that Ubx binds preferentially multiple chromatin sites within single genes, and interacts with the same genes in different tissues but mostly via tissue-specific chromatin sites. In order to identify the origin of these differences, we investigated the chromatin landscape and the Ubx binding motifs within these regions, which revealed that Ubx interacts primarily with non-canonical Hox/Ubx motifs in the mesoderm, while it preferentially uses the classical Hox/Ubx motif for chromatin interactions in the neuronal system. Intriguingly, these novel binding motifs were predicted to have lower binding affinities for Ubx than the classical Hox/Ubx motif. Finally, we also found that the distribution of histone marks at Ubx chromatin sites is different in the mesoderm and the neuronal system

In sum, our analysis of Ubx binding behavior in two tissues identified important differences with regards to underlying sequence patterns, binding affinities and binding site distributions, which we hypothesize to be a consequence of mesoderm development and evolution.

## 2. Results

### 2.1. Ubx Chromatin Interactions Are Highly Specific in Two Different Tissues

Hox proteins are active in most cell types within a segment. On example is the Hox protein Ubx, which is expressed in the ectoderm and nervous system from para-segment 5 to 12 corresponding to the segments T3 to A6, while Ubx expression in the mesoderm is found in A1 to A6 (Figure 1A–C). Intriguingly, Hox proteins perform highly specific functions in the different cell types [4], although they bind rather unspecific DNA sequences with a core sequence of T(A/T)AT(T/G)(A/G) [12,13,18]. Thus, one of the questions in the field is how these two contradictory behaviors can be reconciled. One possibility is that Hox proteins bind different and more specific sequences in different cellular contexts; however, so far, binding preferences of Hox proteins in different cell or tissue types have not been studied in a systematic fashion.

We aimed at closing this gap by using the Hox protein Ultrabithorax (Ubx) and its chromatin interactions in two distinct tissues, the mesoderm and the neuronal system, as a model. To this end, we re-investigated existing Ubx ChIP-Seq profiles retrieved from neuronal and mesodermal cells at two developmental time points: embryonic stages 10–13 (4–9 h after egg lay (AEL), early time point, specification) and embryonic stages 14–17 (10–18 h AEL, late time point, differentiation) [4]. The used reads were generated from chromatin, which was retrieved from nuclei isolated from each tissue separately via the INTACT (isolation of nuclei tagged in specific cell types) method [19]. This method relies on the cell-type specific biotinylating of the nuclei followed by streptavidin-based pull-down. The ChIP was performed with a home-made Ubx antibody, which identified 13,551 Ubx-related mesodermal and 7148 neuronal peaks for the stages 10–13, as well as 9529 mesodermal and 1734 neuronal peaks for the stages 14–17. For the re-analysis, we compared all mapped Ubx binding peaks in the two different tissues and at the different time points using DiffBind [20]. Cross-correlation heatmap analysis revealed that binding of Ubx is highly distinct, as the overlap of binding events in the different tissues or developmental stage was minor (Figure 1D). In the same line, principal component analysis (PCA) of the normalized read counts clearly separated all samples, in particular the early neuronal profile (Figure 1E). Theses result showed that Ubx chromatin interactions in the different tissues and developmental stages are mostly non-overlapping. One example highlighting this differential binding behavior is found in the *midline* (*mid*) locus: Ubx interacts with the mid promoter in both tissues, while the remaining interactions are distinct (Figure 1F).

Taken together, the differential binding analysis revealed that Ubx interacts with different chromatin sites in the mesoderm and the nervous system of *Drosophila* embryos.

### 2.2. Ubx Mostly Interacts with Multiple Chromatin Sites in the Different Tissues

We next wanted to characterize the differential binding behavior of Ubx in more detail. To reduce the complexity of the data, we focused on the stages 10–13 for the subsequent analyses. We first asked how many of the Ubx chromatin interactions occur in the same or different genes in both tissues. To this end, we associated Ubx chromatin peaks to genes, which revealed that 35% of genes were bound by Ubx in both tissues, which will be referred to as “common genes” in the rest of the manuscript (3552 overlapping/common genes). The remaining genes were specifically bound only in the mesoderm (mesoSpec, 29%, 2865 genes) or the neuronal system (neuroSpec, 36%, 3596 genes) (Figure 2A, Appendix A). These differences between tissue specific Ubx peaks and the co-bound genes (common genes) are statistically significant (Figure 2B). In a next step, we asked whether the frequency of Ubx binding to these three categories of genes, common, mesoSpec, and neuroSpec, is different. Thus, we calculated the number of Ubx chromatin peaks occurring in the vicinity of gene bodies. This analysis revealed that genes in the “common” category are in 90% of the cases bound by Ubx multiple times in both tissues, while one Ubx chromatin interaction per gene and tissue occurred in only 10% of the common genes (Figure 2C). Multiple binding events contain two and up to five peaks in most of the cases (Figure 2C’). This relationship is shifted in the mesoSpec and neuroSpec category, as many more genes (35–40%) are only bound once by Ubx in the respective tissue (Figure 2C). The GO term analysis using higher order gene ontology clustering by the WEADE tool [21] indicated that all three categories contain a defined signature: in the mesoderm, Ubx preferentially interacts with genes playing a role in trafficking, transport and the cell cycle, whereas in the neuronal system, it is mostly genes with stem cell related functions that are bound by Ubx (Figure 2D).

Focusing on the “common gene” category, we wondered whether Ubx binding frequency correlates with gene function. To more systematically analyze this relationship, we compared GO term overrepresentations in the common genes separated in single peaks (singlePeaks) and multiple peaks (multiPeaks) using the WEADE tool [21]. We found that many genes coding for TFs, signaling pathways molecules, and trafficking transport proteins are only once bound by Ubx, whereas genes coding for stem cell factors, cytoskeleton and cell adhesion molecules, as well as growth-related proteins, are often bound multiple times (Figure 2D). An example for a gene containing multiple peaks is *teyrha-meyrha* (*tey*), an E3 ubiquitin ligase encoding gene that is multiply bound by Ubx peaks in the mesoderm and neuronal tissue (Figure 2F), and an example for single peak gene is *snail* (*sna*), a transcription factor encoding gene which is bound only once in both tissue contexts (Figure 2G). Intriguingly, differences in Ubx binding frequencies also correlate with chromatin locations, as Ubx chromatin interactions in singlePeaks genes are often at the promoter, while Ubx interactions in multiPeaks genes are primarily found in intronic and intergenic regions (Figure 2E).

In sum, this analysis showed that Ubx interacts at the same time with many genes in the mesoderm as well as the neuronal system, and does so preferentially via multiple and tissue-specific chromatin sites.

### 2.3. Novel Sequence Patterns Resembling Hox Motifs Are Enriched at Ubx Chromatin Sites in the Mesoderm

So far, our analysis has focused on the “common genes” category. In a next step, we wanted to elucidate whether Ubx binding to the same genes in different tissues is of different quality. To this end, we focused on those genes bound by Ubx multiple times in enhancer regions (intronic, intergenic, distal enhancer) in both tissues, and searched for over-represented DNA motifs. To increase specificity of this search, we sub-divided the Ubx peaks in these regions into three categories: (1) Ubx peaks within common genes that are only present in enhancer regions in the mesoderm, meso_ONLYpeaks, (2) Ubx peaks within the common genes that are only present in enhancer regions in the neuronal system, neuro_ONLYpeaks, and (3) Ubx peaks within the common genes that are present in both tissues within a 2 kb region, which we considered to be due to their close distance part of the same regulatory control element and thus bound by Ubx in both tissues, 2kb_common. Interestingly, this categorization of peaks uncovered that the majority of Ubx chromatin peaks (around 75%), despite being present in both tissues in the same genes, are specific to one of the two tissues, and only a smaller proportion was present at the same location in both tissues (Figure 3B).

We next performed motif enrichment analysis in all three categories using MEME [22,23]. We found some known motifs to be over-represented in all three categories, in particular the motif for the ubiquitous factor Trithorax-like (Trl), while other known motifs were found to be over-represented only in a sub-set. For example, the motif for the mesodermal specification factor Tinman (Tin) was over-represented in the meso_ONLYpeaks and the 2kb_common category, while the motif for Intermediate neuroblasts defective (Ind), a TF important for nervous system development, was only over-represented in the neuro_ONLYpeaks category (Figure 3C). Strikingly, although the classical Ubx/Hox DNA motif was found over-represented in the neuro_ONLYpeaks and 2kb_common categories, it was not found among the mesoderm specific Ubx peaks (meso_ONLYpeaks) (Figure 3C). This finding raised the possibility that Ubx interacts with a non-canonical Hox motif in the mesoderm. Thus, we used the STREME (Sensitive, Thorough, Rapid, Enriched Motif Elicitation) sub-routine within the MEME suite to discover novel sequence patterns within the meso_ONLYpeaks category, and identified several novel motifs [24]. Intriguingly, some of them were deviations of the classical Hox/Ubx motif (Appendix A), which could represent so far unknown mesoderm-specific Ubx binding motifs. However, all of them contained at least one -AT- rich region. As a control, we also performed the STREME analysis on the neuro_ONLYpeaks and 2kb_common categories. Although we identified novel motifs, none of them showed a comparable resemblance to the classical Hox motifs to the motifs identified in the meso_ONLYpeaks category (Appendix A). Thus, we assumed that Ubx interacts with regulatory regions via the classical Hox/Ubx motif in the nervous system and in the mesoderm via non-canonical, mesoderm-specific Hox/Ubx motifs. To further confirm these results, we performed the same analysis on Ubx binding regions identified in later stages of development, when muscles and neurons are differentiated. Preliminary results indicated that Ubx-related chromatin binding in the neuronal system is associated with the classical Hox/Ubx binding motif, while this motif cannot be found in mesodermal binding peaks (Appendix A).

After having identified novel sequence patterns within Ubx-bound chromatin regions, we asked whether the known and novel Hox/Ubx motifs are different with respect to predicted affinity or location in the gene loci. Our analysis of peak distribution in regulatory regions revealed that the classical Hox/Ubx motif, which was the one found over-represented among the neuron-specific Ubx chromatin interactions, was located preferentially in intronic regions. In contrast, Ubx binding in the mesoderm is different, as most of the novel, STREME-identified Hox/Ubx binding sequences were more or less evenly distributed in intronic or intergenic (upstream, downstream) regions (Figure 3D). Using the No Reads Left Behind (NRLB) algorithm, we next calculated the affinities of all (known and novel) Ubx/Hox binding sites located within neuro_ONLYpeaks, meso_ONLYpeaks and the 2kb_common regions. In line with our finding that the classical Hox/Ubx motif was enriched in the neuro_ONLYpeaks regions (Figure 3C), we found the predicted affinities of the Hox sites located in these regions to be the highest, while the affinities of the novel Hox/Ubx sites located in the meso_ONLYpeaks regions were estimated to be very low (Figure 3E).

Taken together, the analysis indicated that Ubx primarily binds to the classical Hox motif in the neuronal system, while it interacts with novel, low affinity sequences in the mesoderm.

### 2.4. Tissue-Specific Ubx Sites Show Differences in Histone Mark Distributions

In the next step, we examined whether the genes commonly bound by Ubx in the mesoderm and neuronal system are characterized by other features than differences in the underlying Hox/Ubx sequence motifs. Regulatory regions and their activity are defined by the chromatin status. Thus, we analyzed the distribution of two histone modifications, H3K27ac and H3K27me3, at tissue-specific Ubx peaks located in enhancer regions of gene loci in the two different tissues. Both histone marks are key modifications of the nucleosome component Histone H3, with H3K27me3, represents a mark associated with repressive chromatin [25,26], while H3K27ac is associated with active promoters and enhancers [27,28]. We found that chromatin sites bound by Ubx specifically in the mesoderm, which harbor a H3K27ac mark, also show a high coverage of this chromatin mark in the neuronal system (Figure 4A, Appendix A). GO-term analysis of the genes associated with these regions showed that they function primarily in the neuro-ectodermal tissue (wing disc, appendage, neuron projection morphogenesis). In contrast, mesoderm-specific Ubx chromatin sites, which carry the repressive H3K27me3 mark, harbor neither the active (H3K27ac) nor the repressive (H3K27me3) mark in the neuronal tissue (Figure 4B, Appendix A). Genes associated with these regions were annotated to function primarily in the neuronal system (cell morphogenesis involved in neuron differentiation, locomotion, neuron system development). This is in line with previous findings [4], and indicates that these regions very likely play a role in the repression of alternative fates to ensure proper mesoderm development. Analysis of the chromatin sites bound specifically by Ubx in the neuronal system revealed a different picture: Ubx sites, which harbor a H3K27me3 mark in the nervous system, also did so in the mesoderm (Figure 4D, Appendix A), while nervous system-specific Ubx sites, which carried the H3K27ac mark, were devoid of both histone marks in the mesoderm (Figure 4C, Appendix A).

Taken together, these results showed that the chromatin sites bound by Ubx specifically in one of the two tissues behave differently with regards to two histone marks. Our finding that mesoderm-specific Ubx sites marked by H3K27ac carry the same histone mark in the nervous system was surprising, in particular as the associated genes primarily correlated with epidermal/neuronal tissue development.

## 3. Discussion

The evolutionary conserved Hox proteins perform very specific functions in vivo; however, in vitro, they bind rather generic and frequently occurring DNA sequences. This raises two important questions related to Hox specificity: first, how do different Hox proteins control segment-specific features, and second, how does a single Hox protein regulate the development of different cell and tissue types present in one segment in a specific manner. The first question is related to all Hox proteins and has been extensively studied in the last decades [29,30,31]. However, so far a comprehensive analysis of the tissue-specific function of individual Hox proteins has been missing.

To close this gap, we investigated the binding behavior of the Hox protein Ubx in the *Drosophila* mesoderm and neuronal system. We found that Ubx interacts with highly specific and non-overlapping chromatin sites even when interacting with the same gene locus. Most genes were bound by Ubx at multiple sites located in putative regulatory regions (introns, intergenic regions), while single Ubx binding events were less frequent. Analysis of the Ubx sites identified the classical Hox/Ubx binding motif to be enriched in regions bound by Ubx in the neuronal system, but not the mesoderm. In this tissue, a variety of sequences with similarities to the classical Hox motif were found, which we assume to represent divergent Ubx binding motifs specifically related to the mesoderm. Finally, tissue-specific Ubx sites were not only different in the underlying sequence patterns but also in the distribution of active and repressive histone marks. In sum, these results revealed for the first time that Ubx binding in different tissues is highly specific with regards to the location, sequence patterns and underlying chromatin marks.

One puzzling result was our finding that regions marked as “active” and specifically bound by Ubx in the mesoderm were also in the “active” chromatin configuration in the nervous system, in particular as the genes associated with these regions were predicted to function primarily in the nervous system. This distribution of chromatin marks at tissue-specific Ubx sites indicates that the Hox control of nervous system is more restricted and fixed, as active marks at Ubx sites are related to and exclusive for the tissue, which is not the case in the mesoderm. How could such divergent and more “relaxed” binding behavior of the Hox TF Ubx in the mesoderm be explained? Several aspects related to mesoderm development might be helpful to formulate a hypothesis. First of all, the mesoderm is one of the three germ layers present in Bilateria, while a “true” mesoderm, which arises as a result of gastrulation, is absent in Cnidaria, a group of dipoblastic, radially symmetric organisms [32,33]. Thus, in evolutionary terms, the mesoderm is “younger” and has emerged later than the ectoderm and the endoderm [34,35,36]. In contrast, Hox proteins, which define the identity of body parts in animals, are present in both groups, Bilateria and Cnidaria, as shown for the sea anemone Nematostella [37], highlighting that Hox genes are relatively “old”, and were present before the emergence of a true mesoderm. Thus, the evolution of a new tissue might have required already existing TFs such as the Hox proteins to exploit new binding sequences to enable a clear separation of the old and fixed germ layers from the new and evolving one. In line, we found that Ubx interacts in the mesoderm with many divergent, low-affinity Hox/Ubx binding motifs, while its interaction in the neuronal system is restricted to the classical, high-affinity Hox motif. Intriguingly, low-affinity sites have been shown to be critical for Hox segment/regional specificity [16,17], which might be also the case for Hox tissue specificity. Second, the separation of tissues during animal development is controlled by major changes in the chromatin landscape [38], which could leave “footprints” of the already existing tissue in the newly formed one. For example, the *Drosophila* mesoderm emerges from the neuro-ectoderm and is determined when the predominantly active histone marks (H4K8, H3K18, and H3K27ac) set at gene regulatory regions during the activation of the zygotic gene program are complemented or replaced by repressive histone marks (H3K27me3) [39,40,41]. Thus, the mesoderm might keep “active” histone marks related to nervous system development in the course of its emergence, and rely on “repressive” histone marks to guarantee a clear separation of the mesoderm from the neuro-ectoderm. In line, we found such remnants of neuronal transcriptional activity in the mesoderm, while this was not the case in the nervous system.

Based on these assumptions, we hypothesize that the emergence of a new tissue, the mesoderm, might have required the Hox TFs, which define different body parts and thus need to be active in all tissues, to “relax” their molecular properties, such as their binding behavior and preferences, to enable this major transition from diploblastic to triploblastic organisms, and this change in binding preferences seems to be co-adapted for Hox tissue specificity.

## 4. Materials and Methods

### 4.1. Fly Stocks and Antibody Staining

The *w^1118^* fly line served as a wildtype and embryos were prepared for staining and stained as described in [42], and the following antibodies were used: Rb-Mef2 (1:1500, Gift from H. Nguyen, distributed by K. Domsch), Rat-Elav (1:50, DSHB, ELAV-9F8A9), gp-Ubx (1:500, [4]). The embryos were images with the SP5 Leica Confocal microscope and the images further processed with Fiji [43].

### 4.2. Bioinformatic Analysis and Visualization

The data used in the manuscript was deposit on NCBI Gene Expression Omnibus: GSE121752, and annotated against genome dm6. The data sets contained Ubx ChIP-Seq peaks and histone mark disruptions of H3K27ac and H3K27me3 from the mesoderm and neuronal system, as well as of embryonic stages 10–13 and 14–17. Bioinformatics analysis was performed as described in [4]. The Ubx genomic data was used in a DiffBind [20] analysis under standard setting to determine die different bindings in the two tissues and time frames. The following tools were used for downstream analysis: SAMtools [44], BEDtools (intersect and coverage, [45]), motif search by MEME suite web-tool (Frith et al., 2008), deepTools [46] and visualized with IGV, PANTHER (GO biological function complete, Binomial), Bonferroi correction [47,48,49], Higher order GO-term enrichment [21], R tool [50]: ChIPseeker [51], ChIPpeakAnno [52] and NLRB [53], plots were performed in R [50]. Promoter and enhancer definition is related to [54] and [55]. More details concerning single tools can be found in [4,56].

## Figures and Tables

**Figure 1 jdb-09-00057-f001:**
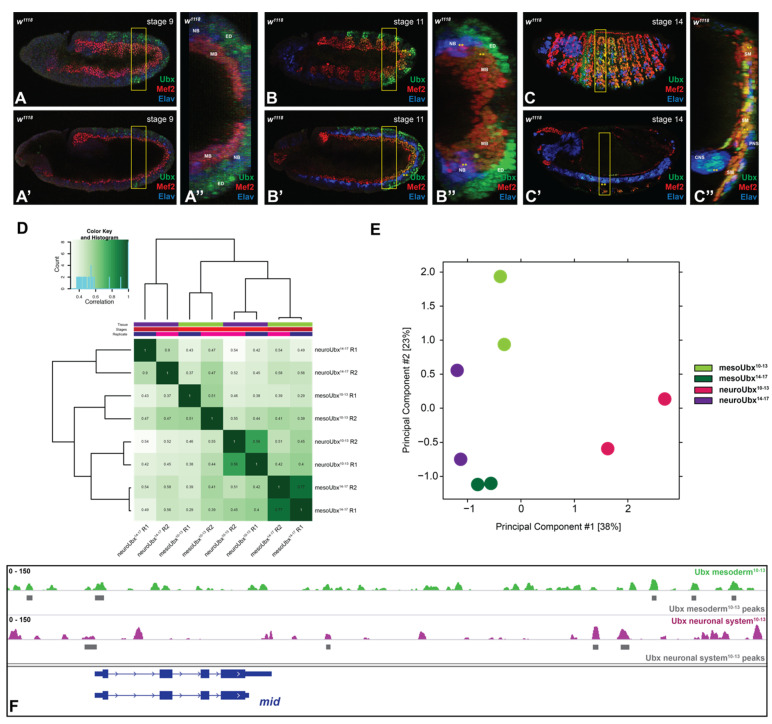
Ubx chromatin interactions in the two tissues are highly specific. (**A–C**) Lateral views of *Drosophila* embryos stained for Ubx (green), the muscle differentiation marker Mef2 (red) and the pan-neuronal marker Elav (blue). (**A**–**C**) Focus is on the mesoderm, (**A’**–**C**’) focus is on the neuronal system. (**A’’**–**C’’**) 3D image reconstruction, region highlighted with the box, illustrates the overlap of Ubx and the tissue-specific markers with **. MB: myoblast, NB: neuroblast, ED: ectoderm, SM: somatic mesoderm, CNS: central nervous system, PNS: peripheral nervous system. Ubx expression is detectable in the ectoderm and very weakly in the nervous system at stage 9. The expression overlaps with tissue-specific markers at stage 11 and 14. (**D**) Cross-correlation heatmap of the called Ubx peaks, indicating a clustering of the replicates and clear separation of the different tissues and time points. (**E**) PCA analysis of the normalized read counts shows that mesodermal and late neuronal samples are more similar than the early neuronal replicates. (**F**) Example showing differential Ubx binding in the midline gene locus (mid). Green: Ubx mesodermal reads stages 10–13 (scale 1–150), purple: Ubx neuronal reads stages 10–13 (scale 1–150), gray boxes: respected accepted peaks, blue: coding region.

**Figure 2 jdb-09-00057-f002:**
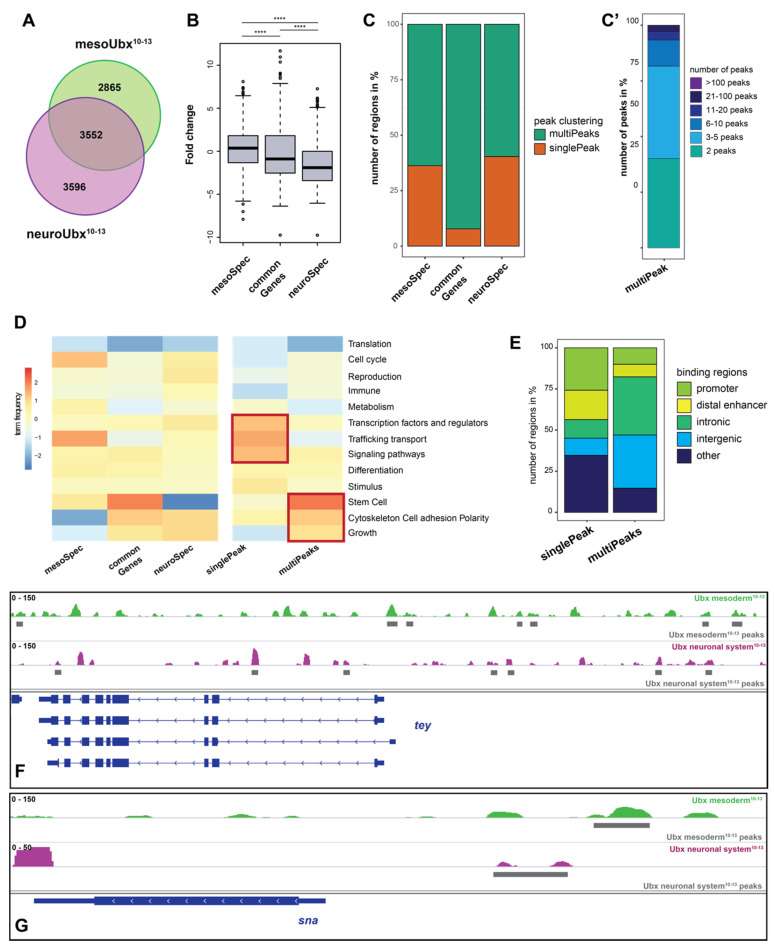
Ubx interacts with multiple chromatin sites in the vicinity of genes. (**A**) Venn diagram of Ubx peaks associated genes in the mesodermal and neuronal tissue, showing a substantial overlap of co-bound and potentially co-regulated genes (common genes). (**B**) Box plot of the DiffBind analysis highlighting the significant difference between tissue specific Ubx peaks and the co-bound genes (common genes). *p*-value **** is < 2.2 × 10^−16^ according to the Wilcox rank test. (**C**) Bar diagram of the peak clustering, indicating that the majority of the common genes contain multiple Ubx peaks (multiPeaks) as compared to the tissue specific peaks, 30–40% single bound genes (singlePeak). (**C’**) Detailed analysis of the common mutiPeaks, showing the frequency of peaks appearing at a single gene (**D**) Heat map of higher order GO-term clustering among the mesoSpec, common and neuroSpec genes. This analysis shows that single peaks are mostly found at genes functionally related to transcription factors, trafficking transport, and signaling pathways, while multiple peaks are found at genes involved in stem cell functions, cytoskeleton cell adhesion polarity and growth. (**E**) Bar diagram of the Ubx peak localization with respect to the gene body. Classified as promoters (−1000 to +10 bp from transcription start site (TSS), 5′ UTR); distal enhancers (−2000 to −1000 bp from TSS, 3′ UTR, downstream); and intron (intronic regions), intergenic (distal intergenic), or other (including exons) regions. The plot shows that multiPeaks are found at enhancer regions, whereas singlePeaks are preferentially associated with promoters. (**F**,**G**) Examples for genes that are bound by Ubx in multiPeaks (tey, teyrha meyrha, F) and with a singlePeak (sna, snail, G). Green: Ubx mesodermal reads stages 10–13 (scale 1–150), purple: Ubx neuronal reads stages 10–13 (F: scale 1–150, G: scale 1–50), gray boxes: respected accepted peaks, blue: coding region.

**Figure 3 jdb-09-00057-f003:**
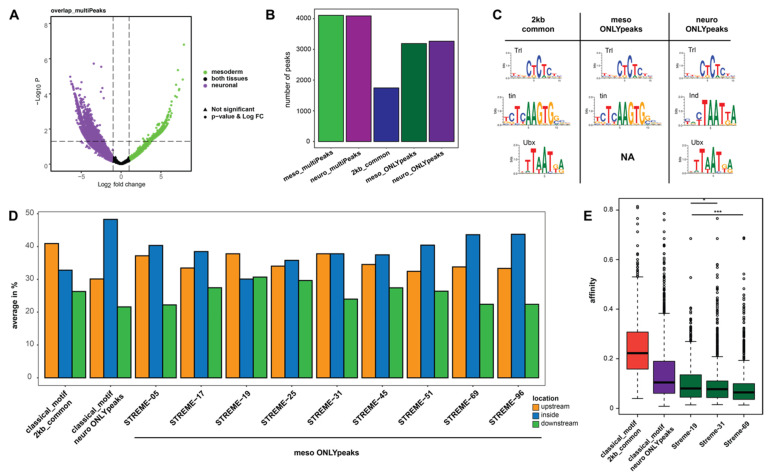
Mesodermal and neuronal Ubx chromatin interactions are different. (**A**) Volcano plot of the selected DiffBind results, the plot shows the distribution of common genes that contain multiPeaks in a tissue-specific manner. (**B**) Bar plot representing the number of peaks tissue specifically associated from the DiffBind analysis used for sub-division in three categories: 2kb_common (mesodermal and neuronal peaks are not as far away as 2 kb and combined in one group), meso_ONLYpeaks, neuro_ONLYpeaks. (**C**) Motif search based on an Analysis of Motif Enrichment (AME within the MEME suit) using the sequences from three categories. The classical Ubx motif is not found in meso_ONLYpeaks. NA: not available. (**D**) Bar plot showing the distribution of classical and novel motifs, which were identified using the Sensitive, Thorough, Rapid, Enriched Motif Elicitation method (STREME) within the MEME suit, in the 2kb_common, neuro_ONLYpeaks, meso_ONLYpeaks category. The plot displays a variance of motif location in meso_ONLYpeaks data set. (**E**) Predicted Ubx binding affinities to selected genomic regions. The classical Hox motif in 2kb_common and neu-ro_ONLYpeaks, as well as selected motifs for the meso_ONLYpeaks data set. The classical motifs show a high affinity, whereas the selected motifs in the meso_ONLY data set have a low affinity. The analysis used model 6: Selex data for Ubx isoform Ia. *p*-value: * is < 0.05 and *** is < 0.001 according to the Wilcox rank test.

**Figure 4 jdb-09-00057-f004:**
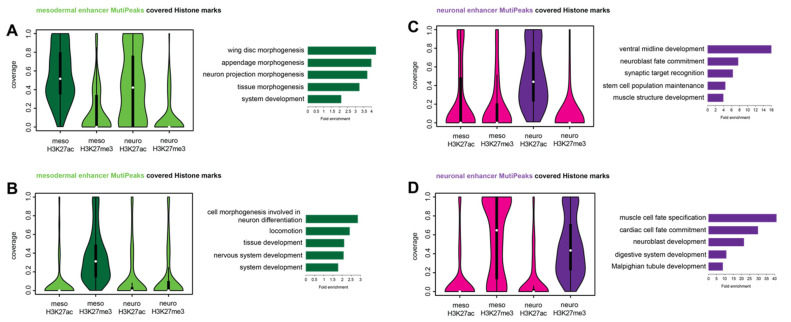
Histone modification at tissue-specific Ubx chromatin sites are different. (**A**,**B**) Ubx bound mesoderm-specific enhancers from the meso_ONLYpeaks data set were analyzed with respect to histone mark distribution. (**A**) The focus is on mesodermal, Ubx-bound regions covered with H3K27ac marks (dark green). These genomic regions are also covered with acetylation marks in the neuronal system (neuro H3K27ac, light green). Bar diagram: GO-term analysis of the selected data (dark green). (**B**) The focus is on mesodermal, Ubx bound regions covered with H3K27me3 marks (dark green), these regions do not contain any other histone mark studied. Bar diagram: GO-term analysis of the selected data (dark green). (**C,D**) Ubx bound neuronal-specific enhancers from the neuro_ONLYpeaks data set were analyzed with respect to their histone mark distribution. (**C**) The focus is on neuronal, Ubx-bound regions covered with H3K27ac marks (dark purple), these regions contain no other mark in a different tissue. Bar diagram: GO-term analysis of the selected data (dark purple). (**D**) The focus is on neuronal, Ubx-bound regions covered with H3K27me3 marks (dark purple), these genomic regions are also covered with methylation marks in the mesoderm (meso H3K27me3, light purple). Bar diagram: GO-term analysis of the selected data (dark purple).

## Data Availability

Sequencing date is available on GEO: GSE121754 (all), GSE121670 (RNA-Seq), GSE121752 (ChIP-Seq).

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
