# Peer review of "An Evolutionary Perspective on Hox Binding Site Preferences in Two Different Tissues"

_jdb, 2021, doi:10.3390/jdb9040057_

Round 1
Reviewer 1 Report
The manuscript entitled "An evolutionary perspective on Hox binding site preferences in two different tissues" by Folkendt et al. reports the use of bioinformatic tools to assess a fundamental question in developmental gene regulation: how a Hox transcription factor, known to recognize Hox-specific low-affinity binding sites, can act with such high specificity in different cell and tissue types? They use as model the drosophile Ubx protein expressed in mesoderm and neuronal system to analyze its binding behavior and compare it to the distribution of two histone marks, H3K27ac and H3K27me3.
Comments:
This is a very interesting piece of work that used existing bioinformatic data to provide a new light on a long-term question. I recommend publication but some concerns need to be answered for clarification.
- Fig. 1D: For the heatmap panel, it would be important to include the correlation factor in each tile even though a scale is provided. This will give more accurate data and probably support better the conclusion reached by the authors.
For instance, according to the color of the tiles, it seems that the correlation between neuroUbx 14-17R2 and mesoUbx 10-13R2 is similar to the correlation existing between mesoUbx 10-13R1 and R2. This goes against the "black & white" conclusion of the authors.
However, if this holds true, the conclusion "the differential binding analysis revealed that Ubx interacts with different chromatin sites in the mesoderm and the nervous system" should be more shaded.
- Fig. 2A: It would be important to repeat the number of genes presented in the Venn diagram in the text.
Fig. 2B: What is the p-value of ****? What was the statistical analysis done to reach this p-value? It should be mentioned in the legend. According to the numbers provided in 2A, it is surprising to see a difference so statistically significant.
Fig. 2G: The scales are different between Ubx mesoderm and Ubx neuronal system but it is difficult to see that since the text is too small in the panel. Please use a bigger font or write it in the figure legend.
- What is exactly an "overlapping gene" for the authors? In Fig. 2F, it is shown that Ubx binds to sites in the tey gene in the mesoderm that are different than the Ubx-binding sites found in the neuronal system. Why not call them "common genes" as Ubx binds to them but at different sites?
- Fig. 3D and E: should be part of Supp. Fig. 1 since the STREME sequences are presented there. Moreover, no explanation about STREME is given in the figure legend of Fig. 3. More explanations should also be provided in the text regarding these sequences.
- Page 4: The sentence "These between tissue specific Ubx peaks and the co-bound genes (overlapping genes) are statistically significant" is incomplete.
- Page 4, last sentence: How can the authors be certain that the binding sites are located in enhancers based on their analyses? More specific marks should be tested and additional experiments should be done to support such assumption. The word "enhancer" should be used with caution in the text.
- Fig. 4A: Why some grey bands (Ubx neuronal system) do not correspond to peaks?
Fig. 4: All the genome views are too small. I suggest to put them in supplementary figures. It will make life more easy for the readers.
Author Response
We would like to thank the reviewer for the constructive criticism and suggestions. Please find below our point-by-point response to the reviewers’ comments and requests. In addition, all major changes are highlighted in red throughout the manuscript.
Reviewer #1: pointed out that the studies is an interesting piece of work, which tries to assess fundamental questions within the field of developmental gene regulation. We addressed the comments of this reviewer in the following way.
- Fig. 1D: For the heatmap panel, it would be important to include the correlation factor in each tile even though a scale is provided. This will give more accurate data and probably support better the conclusion reached by the authors. For instance, according to the color of the tiles, it seems that the correlation between neuroUbx 14-17R2 and mesoUbx 10-13R2 is similar to the correlation existing between mesoUbx 10-13R1 and R2. This goes against the "black & white" conclusion of the authors. However, if this holds true, the conclusion "the differential binding analysis revealed that Ubx interacts with different chromatin sites in the mesoderm and the nervous system" should be more shaded.
- We have added the correlation factors to the heatmap in Fig 1D. In addition, we would like to point out that there is an overlap of binding regions between the different tissues and time points, but this overlap is minor, while differential binding is more prominent.
- Fig. 2A: It would be important to repeat the number of genes presented in the Venn diagram in the text.
Fig. 2B: What is the p-value of ****? What was the statistical analysis done to reach this p-value? It should be mentioned in the legend. According to the numbers provided in 2A, it is surprising to see a difference so statistically significant.
Fig. 2G: The scales are different between Ubx mesoderm and Ubx neuronal system but it is difficult to see that since the text is too small in the panel. Please use a bigger font or write it in the figure legend.
- For Fig 2A: We have added the numbers in the text, which will be referred to as “common genes” in the rest of the manuscript (3552 overlapping/common genes). The remaining genes were specifically bound only in the mesoderm (mesoSpec, 29%, 2865 genes) or the neuronal system (neuroSpec, 36%, 3596 genes) (Figure 2A).”
The statistical analysis in Fig 2B on the different sub groups was performed with the Wilcox rank test. For the test and the Box plot we identified the genomic regions in our DiffBind analysis that are related to the genes in Fig 2A. Despite the “small” number of genes, the differences in genomic regions is highly significant. We provide the information in the figure legend to Fig 2.
We apologies for the small scale in Fig 2G and increased the size and also added the missing information in the figure legend.
- What is exactly an "overlapping gene" for the authors? In Fig. 2F, it is shown that Ubx binds to sites in the the gene in the mesoderm that are different than the Ubx-binding sites found in the neuronal system. Why not call them "common genes" as Ubx binds to them but at different sites?
- We thank the review for the comment and changed the term “overlapping genes” to “common genes”.
- Fig. 3D and E: should be part of Supp. Fig. 1 since the STREME sequences are presented there. Moreover, no explanation about STREME is given in the figure legend of Fig. 3. More explanations should also be provided in the text regarding these sequences.
- At this point we slightly disagree with the reviewer and kept the diagrams of Fig3 D and E in the main figure. We believe that these data should be kept in the main figures, since the diagrams point out that Ubx binding is substantial different in the mesoderm than in the neuronal system.
We added more information regarding the MEME tool STREME in the figure legend to Fig 3.
- Page 4: The sentence "These between tissue specific Ubx peaks and the co-bound genes (overlapping genes) are statistically significant" is incomplete.
- We corrected the sentence in “These differences between tissue specific Ubx peaks and the co-bound genes (common genes) are statistically significant”.
- Page 4, last sentence: How can the authors be certain that the binding sites are located in enhancers based on their analyses? More specific marks should be tested and additional experiments should be done to support such assumption. The word "enhancer" should be used with caution in the text.
- We thank the reviewer for that comment and reviewed the text accordantly. The new version is omitting the word “enhancer”, since we don’t know the exact character of the genomic elements.”… Ubx interacts at the same time with many genes in the mesoderm as well as the neuronal system and does so preferentially via multiple and tissue-specific chromatin sites.”
- Fig. 4A: Why some grey bands (Ubx neuronal system) do not correspond to peaks?
Fig. 4: All the genome views are too small. I suggest to put them in supplementary figures. It will make life more easy for the readers.
- We reviewed the genomic views and corrected the mistakes.
In addition, we moved these genomic views to the new SuppFig 3, as suggest by the reviewer, to make them larger.
Reviewer 2 Report
This paper searches to unravel the mechanism behind Hox specificity in different cell and tissue types. To this end the authors select Ubx and apply bioinformatic approaches to analyze its binding behavior in two different tissues using ChIP-Seq profiles they have previously generated published. I think this re-analysis exposes interesting results. Some concern is regarding the conclusion that Ubx interaction in the mesoderm occurs through divergent-low affinity sites in contrast to the neural tissues. This seems to me overstated and should be moderated and further discussed.
Comments for consideration by the authors:
- Despite their previous publication, it would be desirable to include some description of the Ubx ChIP-Seq profiles used here. How were the tissues separated, the antibody used, number of peaks etc…
- The first part of the study shows that Ubx chromatin interactions are highly stage and tissue specific. For the rest of the study it seems that the authors concentrated only in one stage and this requires justification. Why the differences between stages in the same tissue were not analyzed further?
- The analysis of the overlapping of peaks leads then to consider three categories: overlapping, mesoSpec and neuroSpec. I suggest the authors consider including a table to facilitate visualizations of results, indicating category of genes, number of peaks, location of peaks, GO-terms, etc… The conclusion at the end of page 4 is not clear to me- my understanding is that about one third of Ubx assigned genes are common in both tissues and those common genes are bound in multiple sites (90%) preferentially located in intergenic and intronic regions.
- The analysis of GO terms in overlapping genes is performed according to frequency of peaks. Gene function separates single peaks, preferentially located in promoters, from multiple peaks. I suggest that some reference to numbers is introduced in the text. What is the range of the number of peaks in the 90% of multipeak overlapping genes? Would it be informative to perform functional enrichment in the set of mesoSpec and neuroSpec genes?
- For the quality analysis, it is unclear to me whether the authors concentrated on the overlapping genes and how the three categories were made (meso_ONLYpeaks; neuro_ONLYpeaks and 2kb_common). Some clarification would be desirable.Along the same line, the conclusion at the end of the first paragraph on page 6: “Ubx chromatin peaks … are specific to one of the tissues…” requires further explanation. Have the mesoSpec and neuroSpec genes been analyzed?
- As mentioned, the conclusions reached from the motif search seem overstated: “…we assume that Ubx interacts with enhancers via the classical Hox/Ubx motif in the nervous system and in the mesoderm via non-canonical, mesoderm-specific Hox/Ubx motifs”. As the authors state, this is an assumption, and they should at least discuss on how it could be validated. Could some validation be provided by the analysis of different mesoderm stages? if the divergent, new motifs were also enriched in the mesoderm peaks in other stages. Also, alternative possibilities such as Ubx engaging DNA though interaction with other factors which enriched motifs should be considered
Author Response
We would like to thank the reviewer for the constructive criticism and suggestions. Please find below our point-by-point response to the reviewers’ comments and requests. In addition, all major changes are highlighted in red throughout the manuscript.
Reviewer #2: pointed out that the studies exposes interesting results. We addressed the comments of this reviewer in the following way.
- Despite their previous publication, it would be desirable to include some description of the Ubx ChIP-Seq profiles used here. How were the tissues separated, the antibody used, number of peaks etc…
- We thank the review for the comment and added the information in the text, “The used reads were generated from chromatin derived nuclei, which were isolated from each tissue separately via the INTACT (isolation of nuclei tagged in specific cell types) method (Steiner et al., 2012) that relies on the cell-type specific biotinylating of the nuclei followed by streptavidin-based pull-down. The ChIP was performed with a home-made Ubx antibody, and we identified 13551 Ubx related mesodermal and 7148 neuronal peaks for the stages 10-13, as well as 9529 mesodermal and 1734 neuronal peaks for the stages 14-17.”
- The first part of the study shows that Ubx chromatin interactions are highly stage and tissue specific. For the rest of the study it seems that the authors concentrated only in one stage and this requires justification. Why the differences between stages in the same tissue were not analyzed further?
- We used existing data sets for our bioinformatics analysis and the full data set contains different tissues and time points. We decided to remove the complexity of the data by focusing the in-depth analysis on one stage. “We next wanted to characterize the differential binding behavior of Ubx in more detail. To reduce the complexity of the data, we focused on the stages 10-13 for the following analyses.”
- The analysis of the overlapping of peaks leads then to consider three categories: overlapping, mesoSpec and neuroSpec. I suggest the authors consider including a table to facilitate visualizations of results, indicating category of genes, number of peaks, location of peaks, GO-terms, etc… The conclusion at the end of page 4 is not clear to me- my understanding is that about one third of Ubx assigned genes are common in both tissues and those common genes are bound in multiple sites (90%) preferentially located in intergenic and intronic regions.
- We added a table as Supplementary Table 1, which contains a description of the three categories from Fig 2 regarding different gene categories, GO-terms and peak numbers as well as location.
We also reviewed the conclusion on page 4 and changed it into:” In sum, this analysis showed that Ubx interacts at the same time with many genes in the mesoderm as well as the neuronal system and does so preferentially via multiple and tissue-specific chromatin sites.”
- The analysis of GO terms in overlapping genes is performed according to frequency of peaks. Gene function separates single peaks, preferentially located in promoters, from multiple peaks. I suggest that some reference to numbers is introduced in the text. What is the range of the number of peaks in the 90% of multipeak overlapping genes? Would it be informative to perform functional enrichment in the set of mesoSpec and neuroSpec genes?
- We performed an additional analysis to investigate the frequency of Ubx interacting with the genome on the 90% of multipeaks. Ubx interacts most of the time 2 to 5 times with the genome in the vicinity of one gene, but interactions with more than 100 peaks associated to one gene can also be recorded. The results are illustrated in Fig 2C’. In addition, we also performed the WEADE enrichment analysis for the mesoSpec and neuroSpec genes, these results are included in the Fig 2D. The results show that each data set is different and has its unique signature.
- For the quality analysis, it is unclear to me whether the authors concentrated on the overlapping genes and how the three categories were made (meso_ONLYpeaks; neuro_ONLYpeaks and 2kb_common). Some clarification would be desirable. Along the same line, the conclusion at the end of the first paragraph on page 6: “Ubx chromatin peaks … are specific to one of the tissues…” requires further explanation. Have the mesoSpec and neuroSpec genes been analyzed?
- We aimed to analyze the specificity in the Ubx binding behavior related to the different tissues and focused on genes that contain Ubx peaks in the mesoderm as well as in the neuronal system. Using DiffBind on this selection, we found that even within this pool of peaks some are more related to the neuronal and others to the mesodermal tissue. To include the difference in the analysis we generated the two-tissue specific groups in the common genes, meso_ONLY and neuro_ONLYpeaks. Further on, some peaks are so close to each other that they should not be separated, and it would be better to consider them as one regulatory element, 2kb_common. In the text: ” To increase specificity of this search, we sub-divided the Ubx peaks in these regions into three categories: 1) Ubx peaks within common genes that are only present in enhancer regions in the mesoderm, meso_ONLYpeaks, 2) Ubx peaks within the common genes that are only present in enhancer regions in the neuronal system, neuro_ONLYpeaks, and 3) Ubx peaks within the common genes that are present in both tissues within a 2kb region, which we considered to be due to their close distance part of the same regulatory control element and thus bound by Ubx in both tissues, 2kb_common.
Our focus was on the common genes, but we performed some additional analysis on mesoSpec and neuroSpec genes, the results are added in Fig 2D.
- As mentioned, the conclusions reached from the motif search seem overstated: “…we assume that Ubx interacts with enhancers via the classical Hox/Ubx motif in the nervous system and in the mesoderm via non-canonical, mesoderm-specific Hox/Ubx motifs”. As the authors state, this is an assumption, and they should at least discuss on how it could be validated. Could some validation be provided by the analysis of different mesoderm stages? if the divergent, new motifs were also enriched in the mesoderm peaks in other stages. Also, alternative possibilities such as Ubx engaging DNA though interaction with other factors which enriched motifs should be considered
- We thank the reviewer for the comment and analyzed the later data set. The results are illustrated in the new SuppFig 2. The results show that most of the genes bound by neuronal Ubx are also bound by mesodermal Ubx. Next we also divided the data set in the three categories meso_ONLYpeaks, neuro_ONLYpeaks and 2_kb common, but since 2_kb common and neuro_ONLY behaved the same in respect to the known motif enrichments analysis we used only the tissue specific data set. The results show that the classical Ubx motif can only be found in the neuro_ONLY data set and not in the mesoderm, supporting the results from the early stages.
Round 2
Reviewer 1 Report
This is a revised version of the manuscript entitled "An evolutionary perspective on Hox binding site preferences in two different tissues" by Folkendt et al. In the revised version, the authors have addressed most of the criticisms and the text was clarified. This manuscript can now be considered for publication.
Reviewer 2 Report
The authors have appropriately addressed my comments. I have no further request.